# Toxicological Evaluation of SiO_2_ Nanoparticles by Zebrafish Embryo Toxicity Test

**DOI:** 10.3390/ijms20040882

**Published:** 2019-02-18

**Authors:** Sandra Vranic, Yasuhito Shimada, Sahoko Ichihara, Masayuki Kimata, Wenting Wu, Toshio Tanaka, Sonja Boland, Lang Tran, Gaku Ichihara

**Affiliations:** 1Department of Occupational and Environmental Health, Nagoya University Graduate School of Medicine, Nagoya 466-8560, Japan; sandra.vranic@manchester.ac.uk (S.V.); wendywu1206@163.com (W.W.); gak@rs.tus.ac.jp (G.I.); 2Department of Occupational and Environmental Medicine, Faculty of Pharmaceutical Sciences, Tokyo University of Sciences, Noda 278-8510, Japan; 3Department of Integrative Pharmacology, Mie University Graduate School of Medicine, Tsu 514-8572, Japan; shimada.yasuhito@mie-u.ac.jp; 4Mie University Zebrafish Drug Screening Center, Tsu 514-8572, Japan; 5Department of Environmental and Preventive Medicine, Jichi Medical University School of Medicine, Shimotsuke 329-0498, Japan; 6Department of Human Functional Genomics, Life Science Research Center, Mie University, Tsu 514-8572, Japan; mk.lucky-seven@ezweb.ne.jp; 7Department of Systems Pharmacology, Mie University Graduate School of Medicine, Tsu 514-8572, Japan; tanaka@doc.medic.mie-u.ac.jp; 8Unit of Functional and Adaptive Biology (BFA), Laboratory of Molecular and Cellular Responses to Xenobiotics, CNRS UMR 8251, Universite Paris Diderot, Sorbonne Paris Cite, 75013 Paris, France; boland@univ-paris-diderot.fr; 9Institute of Occupational Medicine, Research Avenue North, Riccarton, Edinburgh EH14 4AP, UK; lang.tran@iom-world.org

**Keywords:** silica nanoparticles, surface functionalization, zebrafish, embryo acute toxicity test, vascularization, bio-distribution, dechorionation

## Abstract

As the use of nanoparticles (NPs) is increasing, the potential toxicity and behavior of NPs in living systems need to be better understood. Our goal was to evaluate the developmental toxicity and bio-distribution of two different sizes of fluorescently-labeled SiO_2_ NPs, 25 and 115 nm, with neutral surface charge or with different surface functionalization, rendering them positively or negatively charged, in order to predict the effect of NPs in humans. We performed a zebrafish embryo toxicity test (ZFET) by exposing the embryos to SiO_2_ NPs starting from six hours post fertilization (hpf). Survival rate, hatching time, and gross morphological changes were assessed at 12, 24, 36, 48, 60, and 72 hpf. We evaluated the effect of NPs on angiogenesis by counting the number of sub-intestinal vessels between the second and seventh intersegmental vessels and gene expression analysis of vascular endothelial growth factor (VEGF) and VEGF receptors at 72 hpf. SiO_2_ NPs did not show any adverse effects on survival rate, hatching time, gross morphology, or physiological angiogenesis. We found that SiO_2_ NPs were trapped by the chorion up until to the hatching stage. After chemical removal of the chorion (dechorionation), positively surface-charged SiO_2_ NPs (25 nm) significantly reduced the survival rate of the fish compared to the control group. These results indicate that zebrafish chorion acts as a physical barrier against SiO_2_ NPs, and removing the chorions in ZFET might be necessary for evaluation of toxicity of NPs.

## 1. Introduction

Nanotechnologies are emerging technologies that have been gaining popularity in the last few decades and are still being developed. These technologies foresee nanoparticles (NPs) for many applications in the fields of medicine, microelectronics, catalysis, cosmetics, drug delivery, and imaging [1,2]. However, human exposure to nanomaterials is a concern since there might be a negative impact on health, not only in occupational settings where NPs are produced and used, but also in the general population and environment considering the life cycle of NPs from manufacturing to recycling and final disposal [3,4]. Due to their small size, which is in the size range of various cellular structures, NPs are capable of interacting with these structures, entering the cells, and/or translocating from the site of exposure toward the circulation and secondary organs [5,6]. NPs can impair cellular functions and be cytotoxic [7,8]. For that reason, it is of crucial importance to understand the key aspects of interactions of NPs with living systems and to properly compare benefits and potential hazard coming from use of nanomaterials.

Silicon dioxide (SiO_2_) NPs have numerous applications, such as additives to drugs, cosmetics, printer toners, varnishes, and food, due to their anti-agglomerations properties [9]. Their characteristics have led to the applications in cancer therapy, DNA delivery, and drug delivery [10,11,12]. The wide use of SiO_2_ NPs and their presence in general and in the occupational environment prioritize the study on their health and environmental effects and the analysis of their potential toxicity and underlying mechanisms, in order to offer essential information on in vivo behavior of SiO_2_ NPs as well as induced toxic responses.

Rodents are widely used to test the potential developmental toxicity of chemical substances, including NPs. The procedures for animal testing and research are highly regulated and the necessity of numerous animals is ethically discussed. Although reliable data for extrapolating toxicant effects to humans are obtained through rodent studies, these experiments are expensive, time consuming, and more restricted by ethics and law. Since genes, receptors, and molecular processes are highly conserved across animal phyla, zebrafish (*Danio rerio*) has been employed as an experimental tool in toxicology for testing chemicals and NPs as an alternative to rodents [13]. In addition, because of their transparent body wall at their younger age, it is easy to visualize and evaluate internal organ toxicities by using transgenic zebrafish with tissue-specific enhanced green fluorescent protein (EGFP) expression. Using vascular-EGFP strains, *Tg* (*fli1:egfp*)/*nacre* zebrafish, we previously investigated the effects of metal oxide NPs on angiogenesis [14].

In the present study, we investigated the effects of fluorescently-labeled SiO_2_ NPs of two different sizes: 25 and 115 nm, with a neutrally charged, pristine surface or with different surface functionalization, rendering them positively or negatively charged using the zebrafish embryo toxicity test method (ZFET). We also evaluated the effects of surface charge and size of SiO_2_ NPs on bio-distribution and vascularization in the embryos of vascular-EGFP zebrafish using fluorescent-labelled NPs.

## 2. Results and Discussion

### 2.1. Characterization of NPs in Suspension

In the present study, the in vivo effects of SiO_2_ NPs with three different surface charges were studied. We used rhodamine-labeled SiO_2_NPs 25 and 115 nm in diameter with hydroxyl function on the surface (pristine, neutral surface charge; N) or functionalized with amino (positive surface charge; +q) or carboxyl groups (negative surface charge; −q). The intensity-weighted hydrodynamic average diameters of dispersed NPs were determined by the dynamic light scattering (DLS) technology. Table 1 shows the mean hydrodynamic diameters and polydispersity index (PdI) of SiO_2_ NPs in embryo culture medium. The hydrodynamic average diameters of amino-, carboxyl-, and hydroxyl-modified 25 nm SiO_2_ NPs were 171.3 ± 8.58, 166.8 ± 2.05, and 132.2 ± 6.99 nm, respectively (Table 1), indicating that NPs were aggregated or agglomerated in the medium. Although DLS data provided the mean hydrodynamic diameters of >100 nm, the presence of nano-sized particles was confirmed in the medium (Table 1). The average hydrodynamic diameters of 115-nm SiO_2_ NPs were 228.9 ± 5.06, 152.7 ± 1.19, and 127.2 ± 1.10 nm for amino-, carboxyl-, and hydroxyl-modified, NPs respectively (Table 1). Hydrodynamic sizes of all NPs dispersed in the embryo medium did not change after 24 h.

SiO_2_ NPs were aggregated or agglomerated in the medium (Table 1). The aggregation state of the NPs influenced the final toxicological outcome. Note that we performed the series of experiments using NPs dispersed in Danieau’s solution supplemented with albumin (0.012 mg/mL) in order to obtain a better dispersion of NPs. However, this experimental set-up (presence of albumin in Danieau’s solution) was toxic for the embryos on its own without any presence of NPs. Moreover, the dispersibility of the NPs was not much improved. Therefore, we decided to continue the treatment of the embryos using the media that are the most suitable for their normal growth and development.

### 2.2. Effects of SiO_2_ NPs on the Development of Zebrafish Embryo

The zebrafish embryo toxicity test (ZFET) has emerged as an alternative in vivo approach to assessing developmental toxicity [15], with some variations including use of tissue-specific EGFP fish lines alone [16] or in combination with behavior assays [17,18]. There are several variations in ZFET protocols for assessment of targeted toxicity though OECD-defined guidelines for ZFET in 2013 [19].

To evaluate the developmental toxicity of SiO_2_ NPs, the zebrafish embryos were exposed to SiO_2_ NPs with different surface charges of two different sizes, 25 and 115 nm, starting from 6 hpf. Survival rate, hatching rate, and gross morphological changes were examined at each time point of 12, 24, 36, 48, 60, and 72 hpf after exposure to SiO_2_ NPs at 3.125, 6.25, 12.5, 25, 50, and 100 mg/L (Appendix A). The survival rate (Figure 1a) and hatching rate (Figure 1b) at 48 and 72 hpf were not affected by exposure to all types of SiO_2_ NPs at the highest concentration (100 mg/L). Correspondingly, there were no obvious morphological changes at 72 hpf in zebrafish exposed to all types of SiO_2_ NPs at the highest concentration (100 mg/L) (Figure 2a,b). These results are partially in accordance with the previous study that showed no toxicity in the early life stage of zebrafish exposed to core-shell silica NPs [20]. Although Li et al. [21] previously reported that exposure to smaller SiO_2_ NPs, whose diameter was less than 50 nm, at 300–1000 mg/L caused Parkinson’s-like behavior in adult zebrafish. No locomotor disturbance was observed at 72 hpf in zebrafish exposed to 100 mg/L for all types of SiO_2_ NPs in the present study.

### 2.3. Effect on Angiogenesis in Zebrafish Embryos

To evaluate the potential toxicity on angiogenesis, the number of sub-intestinal vessels running transversally between planes including the second and seventh intersegmental vessels was counted in 72 hpf embryos. The result showed that there were no significant differences in the number of transversely-running sub-intestinal vessels in zebrafish exposed to all types of SiO_2_ NPs at the concentration of 100 mg/L. Previously, we demonstrated that exposure to CuO NPs reduced the number of transversely-running sub-intestinal vessels in the same TG zebrafish at 5 dpf and down-regulated the expression of VEGF and VEGF receptors in endothelial cells sorted by the Fluorescence Activated Cell Sorter (FACS) [14]. Brundo et al. [22] showed that Au NPs exposure affected the expression of biomarkers, such as metallothionein, in zebrafish embryos even with normal survival rate and phenotype. Therefore, we analyzed the gene expression of biomarkers involved in angiogenesis (*vegfa* and its receptors: *flt-1* and *kdr*). We found that there were no significant changes in their expression in zebrafish embryos exposed to all types of NPs (Figure 3a–c). These results indicate that SiO_2_ NPs have no effects on angiogenesis in all size and types of surface-charges in regular ZFET.

The SD values of the gene expression of VEGF and its receptors are relatively high in the present study. However, these variations are possible because another study [23] that investigated the expression of VEGF in zebrafish embryos similarly reported high SD values in their data. Total RNA was extracted from the whole body of zebrafish and was subjected to quantitative PCR in both studies. This might explain how the variations in gene expression of VEGF and its receptors were high. If the expression of VEGF and its receptors had been analyzed using total RNA isolated from the sorted green fluorescent protein (GFP)-positive endothelial cells, the variations might have been minimal. Further studies are needed to evaluate SiO_2_ NPs effect on angiogenesis and the gene expression associated with angiogenesis.

### 2.4. Protective Effect of Chorion Against SiO_2_ NPs

Contrary to the present ZFET results, several reports showed that SiO_2_ NPs exert acute toxicity in vitro (immortalized mammalian cell lines) and in vivo (mouse and rat) [24,25]. Physico-chemical properties, such as size, surface area, and surface properties including charges, were found to play a key role in the toxicity of SiO_2_ NPs [26]. To better understand the non-toxic effect of SiO_2_ NPs in ZFET, we examined the localization of NPs using rhodamine-labelled SiO_2_ NPs in the zebrafish embryos. As shown in Figure 4, all types of SiO_2_ NPs accumulated on the surface of chorion before hatching. Even after hatching, the NPs were not detected inside the fish bodies (Figure 5).

To evaluate the protective effect of chorion against SiO_2_ NPs, we exposed dechorionated (chorion removed) embryos to SiO_2_ NPs. After chemical removal of the chorion at 24 hpf, survival rate was not affected and gross morphological changes were not observed in the embryos exposed to 100 mg/L of SiO_2_ NPs at 48 and 72 hpf. Accordingly, the chorion was carefully removed at 6 hpf and the embryos were exposed to different surface-charged SiO_2_ NPs. As a result, exposure to positively surface-charged SiO_2_ NPs (25 nm) significantly (*p* < 0.05) reduced the survival rate of zebrafish at 24 and 48 hpf (Figure 6). It has been shown that positively charged NPs showed a higher cell uptake rate owing to its electrostatic interaction with cells [27,28]. Since the cell membrane is negatively charged in general, reduction in the survival rate of zebrafish by positive surface-charged NPs might be due to the higher uptake of NPs into cells, resulting from interaction of positively charged NPs and negatively charged cell membrane. However, further studies are needed to test this hypothesis.

In rodents, several kinds of metal-bearing NPs exhibit reproductive [29] and neurodevelopmental toxicities [30], probably due to oxidative stress, inflammation, or DNA damage [31]. Apoptosis induced by DNA damage and/or oxidative stress might be a mechanism underlying the developmental toxicity of SiO_2_ NPs in zebrafish embryos, as shown in other studies on NP toxicity [32,33,34]. In addition, our results demonstrated that all types of SiO_2_ NPs, regardless of the size or charges, were completely trapped by chorion, and subsequently blocked from exhibiting the toxicity to zebrafish embryos (Figure 5). Chorion might adsorb NPs as a natural barrier, as suggested by other studies [35,36]. The chorion possesses canals whose pore size is approximately 0.6–0.7 µm [37], which is larger than the size of the NPs in the present study, suggesting that the barrier property of chorion against NP transport is explained by adsorption of NPs by the chorion.

In the preset study, we focused on the effects of SiO_2_ NPs on the development of zebrafish and the effects of SiO_2_ NPs on angiogenesis. Because aorta and large vessels are formed by 48 hpf and small vessels in the abdomen, such as sub-intestinal vessels, are formed by around 72 hpf in zebrafish [38], we exposed SiO_2_ NPs at 6 hpf and observed the effect of SiO_2_ NPs on the formation of blood vessel at 72 hpf. The evaluation of the toxicity of SiO_2_ NPs by the exposure to SiO_2_ NPs after the chorions disappear at 48 hpf is also important. Further investigation is needed to determine the effects of SiO_2_ NPs after the absence of chorion. The effects of NPs on young zebrafish (after 8 dpf) should also be evaluated because the larvae become independent and start oral feeding after the nutrients contained in the yolk sac are exhausted. This topic can be considered as an essential theme for further investigation.

## 3. Materials and Methods

### 3.1. Preparation and Characterization of NPs Suspensions

Rhodamine-labeled silica NPs, 25 and 115 nm in diameter, with hydroxyl groups on the surface (neutral surface charge; N) or functionalized with amino modified (positive surface charge; +q) or carboxyl modified (negative surface charge; –q) were purchased from HiQ-Nano, Arnesano, Italy. All NPs were suspended in water at 25 mg/L and then dispersed with 0.3× Danieau’s solution (17.4 mM NaCl, 0.21 mM KCl, 0.18 mM Ca(NO_3_)_2_, 0.12 mM MgSO_4_, and 1.5 mM HEPES buffer, pH 7.6). The particle size distribution was determined with a nano-zetasizer (Zetasizer Nano S; Malvern Instruments, Worcestershire, UK), which uses DLS technique [39]. Hydrodynamic sizes of all NPs dispersed in the embryo medium at the highest concentration used for the treatment were analyzed from the time of exposure up to 24 h in order to check the stability of the NP solutions during the contact with embryos. The fluorescence intensities of three types of silica NPs were measured at different concentrations using an ARVO™MX 1420 Multilabel Counter (Perkin Elmer, Waltham, MA, USA). The successive dilutions of the medium at particle concentrations of 100, 50, 25, 12.5, 6.25, and 3.125 mg/L were completed immediately prior to exposure.

### 3.2. Zebrafish

Zebrafish were maintained in the facility at Mie University according to the standard operational guidelines. The nacre/fli1:egfp zebrafish with transparency, which facilitates in vivo monitoring of vascularization, were obtained by cross-breeding nacre mutants [40] and fli1:egfp transgenic zebrafish [41]. The zebrafish were acclimated to experimental conditions (28 ± 0.5 °C; light:dark/14:10 h; daily water change), as described previously [42]. All animal procedures were conducted according to the Japanese Animal Welfare Regulation Act and Management of Animals (Ministry of Environment of Japan) and complied with international guidelines. Ethical approval from the local Institutional Animal Care and Use Committee was not sought, since this law does not mandate the protection of fish. 

### 3.3. Zebrafish Embryo Toxicity Test (ZFET)

Zebrafish embryo toxicity test (ZFET) for SiO_2_ NPs was performed according to the Organisation for Economic Co-operation and Development (OECD) guideline [19]. The day before egg collection, a single female and two male nacre/fli1:egfp zebrafish were placed in mating tanks. The next morning, mating was initiated by light stimuli and the fertilized eggs were collected. These eggs were incubated in 0.3× Danieau’s solution in a 6-well plate at the concentration of 20 eggs/well in 3 mL medium at 28 °C. The good quality eggs that were beginning to develop a yolk sac and animal pore were chosen. At 6 h post fertilization (hpf), the embryos were exposed by immersion in 0.3× Danieau’s solution or the same solution containing SiO_2_ NPs up to 72 hpf. Embryo medium was changed every 24 h with a new solution of NPs. Fluorescent image acquisition was performed using a Leica MZ16F stereoscopic microscope (Leica Microsystems, Wetzlar, Germany) equipped with DP71 digital camera (Olympus, Tokyo, Japan).

### 3.4. Evaluation of Survival Rate, Hatching Rate, and Gross Morphological Changes

Each of the 20 embryos were incubated in a 6-well plate with 3 mL of embryo medium. Then, the embryos were exposed to 3.125, 6.25, 12.5, 25, and 50 mg/L or 6.25, 12.5, 25, 50, and 100 mg/L of rhodamine-labeled silica NPs 25 or 115 nm in diameter from 6 to 72 hpf. Survival rate, hatching time, and gross morphological changes were examined at each time point: 12, 24, 36, 48, 60, and 72 hpf. The number of dead embryos/fish, the number of non-hatched or hatched embryos, and the number of deformed embryos/fish were recorded. Using MZ16F stereotypic microscope (Leica Microsystems, Wetzlar, Germany), the embryos were first observed using bright field and then fluorescence, and the photos were captured at the indicated time points.

### 3.5. Evaluation of NPs Localization

The embryos were anesthetized using 100 ppm 2-phenoxyethanol (ethylene glycol monophenylether) in 0.3× Danieau’s solution in the wells and mounted on the slides. The embryos were incubated in the embryo medium for at least 15–20 min for rinsing to remove NPs before anesthesia. To check the localization of the rhodamine-labeled SiO_2_ NPs in the embryos, live imaging of embryos after exposure was performed using MZ16F stereotypic microscope (Leica Microsystems) with a DP71 digital camera (Olympus, Tokyo, Japan) at 24, 48, and 72 hpf.

### 3.6. Evaluation of Development of Subintestinal Vessels and Measurement of Gene Expression of Vascular Endothelial Growth Factor (VEGF) and VEGF Receptors

After 72 hpf, zebrafish were transferred to 96-well microplates and observed for gross morphological changes under a fluorescent stereomicroscope. The number of transversely-running subintestinal vessels (perpendicular to the anterior-posterior axis) between the transverse planes containing the 2nd and 7th intersegmental vessels, was counted. Total RNA was extracted from pooled embryos using the RNAqueous-MicroTotal RNA Isolation kit (Thermo Fisher Scientific, Waltham, MA, USA) according to the manufacturer’s instructions. The concentration of total RNA was quantified by spectrophotometry (ND-1000; NanoDrop Technologies, Wilmington, DE, USA). The first-strand cDNA was synthesized from 200 ng of total RNA using the SuperScript III cDNA Synthesis Kit (Thermo Fisher Scientific) with oligo dT primers (Thermo Fisher Scientific). cDNA (*n* = 5 in each group) was subjected to quantitative polymerase chain reaction (PCR) analysis with FastStart Universal Probe Master Mix (Roche, Basel, Switzerland) and primers for VEGFa and VEGF receptors (VEGFR1; Flt-1: Fms-like tyrosine kinase and VEGFR2; KDR: kinase insert domain receptor) using an ABI 7300 Real-Time PCR system (Thermo Fisher Scientific), as described previously [14].The gene expression level was normalized to that of β-actin in the same cDNA.

### 3.7. Evaluation of Survival Rate and Gross Morphological Changes after Dechorionation of Zebrafish Embryos

To evaluate the protective effect of chorion against SiO_2_ NPs, we exposed SiO_2_ NPs to dechorionated (chorion removed) embryos. For chorion removal (dechorionation), 6 or 24 hpf embryos were immersed in 1.5 mg/mL of pronase (Roche Molecular Systems, Pleasanton, CA, USA) and the chorion was carefully removed according to the standard protocol [43]. After chemical removal of chorion at 6 hpf, the embryos were exposed to 100 mg/L of rhodamine-labeled SiO_2_ NPs at the same time (6 hpf). Survival rate and gross morphological changes were examined at 24 and 48 hpf. After chemical removal of chorion at 24 hpf, the embryos were exposed to the same dose of SiO_2_ NPs from 6 hpf, and survival rate and gross morphological changes were examined at 48 and 72 hpf.

### 3.8. Statistical Analysis

All parameters were expressed as mean ± standard deviation (SD). Statistical analyses were performed using one-way analysis of variance (ANOVA) followed by Dunnett’s post hoc test. A *p* value less than 0.05 was considered statistically significant.

## 4. Conclusions

The toxicity of SiO_2_ NPs has been studied for many years and is linked with chronic bronchitis, emphysema, and silicosis in mammals [44,45,46]. For evaluation of developmental toxicity of SiO_2_ NPs, the zebrafish embryo is a powerful tool as an alternative model of rodents and as a suitable model for rapid growth and live imaging of the internal organs. However, OECD-approved ZFET protocol typically employs zebrafish embryos with intact chorions, which can obstruct NPs uptake independently from the size and surface charges. Our results suggest that chorion removal (dechorionation) in 6 hpf embryos might be necessary to determine the toxicological impact of NPs.

## Figures and Tables

**Figure 1 ijms-20-00882-f001:**
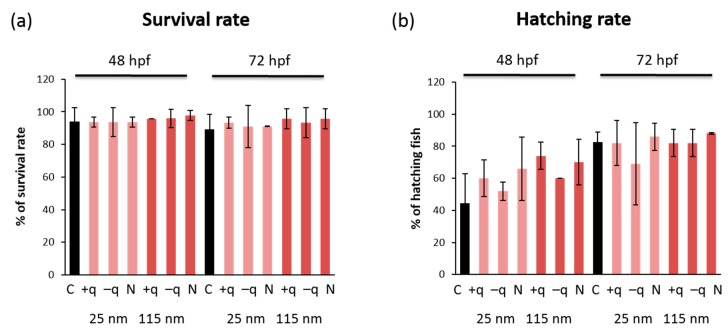
The effect of exposure to SiO_2_ NPs on the development of zebrafish embryos. (**a**) Survival rate and (**b**) hatching rate after exposure to 100 mg/L of 25- or 115-nm SiO_2_ NPs with different surface charges at 48 and 72 hpf. Data are represented as mean ± SD (standard deviation).

**Figure 2 ijms-20-00882-f002:**
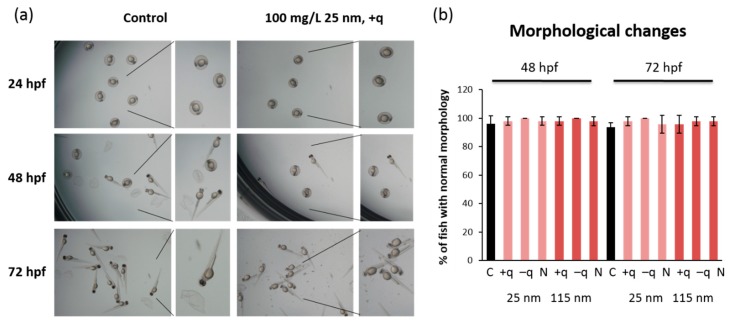
The effects of SiO2 NPs on gross morphological features of zebrafish embryos. (**a**) Representative images of embryos exposed to positively charged SiO_2_ NPs (25 nm) at 100 mg/L and (**b**) gross morphological changes after the exposure to 100 mg/L of 25 or 115 nm SiO_2_ NPs at 48 and 72 hpf. Data are represented as mean ± SD.

**Figure 3 ijms-20-00882-f003:**
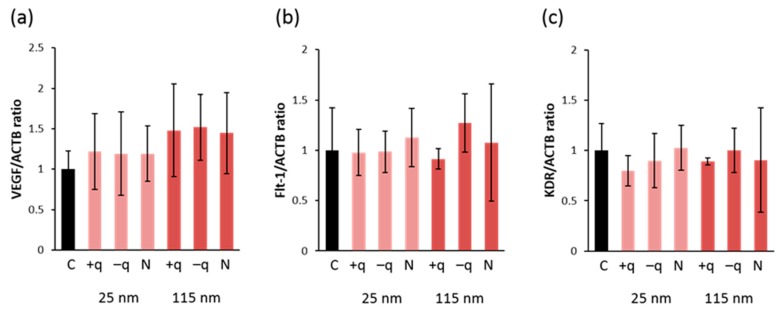
Gene expression of VEGFa and VEGF receptors in zebrafish. Expression levels of (**a**) VEGFa and VEGF receptors: (**b**) Flt-1 and (**c**) KDR in embryos exposed to 100 mg/L of 25 or 115 nm SiO_2_ NPs with different surface-charge at 48 and 72 hpf. The mRNA levels were normalized by the amount of β-action mRNA. Data are represented as mean ± SD.

**Figure 4 ijms-20-00882-f004:**
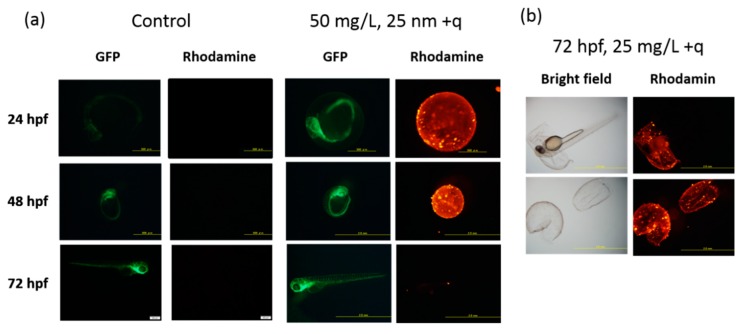
Representative fluorescence microscopy images of zebrafish embryos. (**a**) Images of embryos exposed to 0 (control) or 50 mg/L of positively charged SiO_2_ NPs (25 nm) at 24, 48, and 72 hpf and (**b**) embryos exposed to 25 mg/L of positively charged SiO_2_ NPs (25 nm) at 72 hpf.

**Figure 5 ijms-20-00882-f005:**
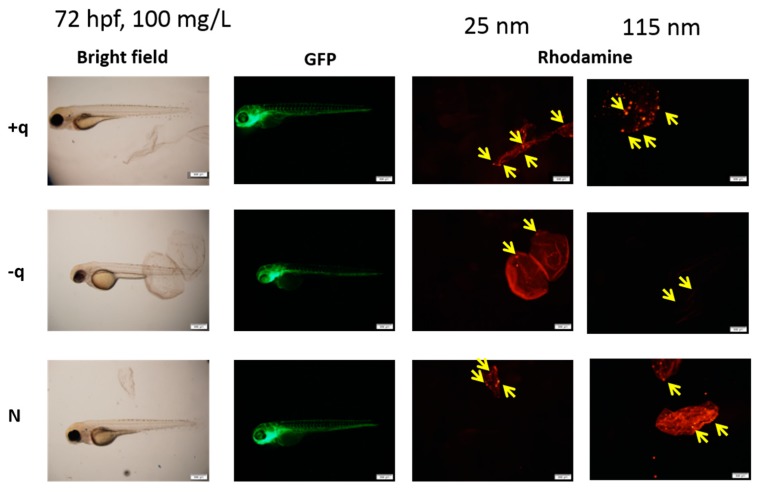
Representative fluorescence microscopy images of embryos exposed to 100 mg/L of 25 or 115 nm SiO_2_ NPs with different surface charges at 72 hpf.

**Figure 6 ijms-20-00882-f006:**
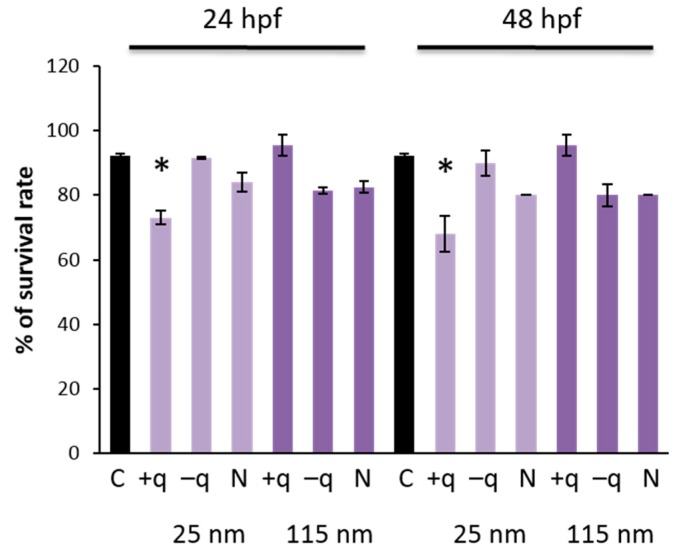
Survival rate at 24 and 48 hpf in zebrafish embryos exposed to 100 mg/L of 25 or 115 nm SiO_2_ NPs with different surface charges after chemical removal of chorion at 6 hpf. Data are represented as mean ± SD. * *p* < 0.05 compared with the control (ANOVA followed by Dunnett’s multiple comparison test).

**Table 1 ijms-20-00882-t001:** Characterization of nanoparticles (NPs).

NPs	Hydrodynamic Size (nm)	PdI	Intensity of Particles of less than 100 nm (%)	Volume of Particles of less than 100 nm (%)
25 nm +q	171.3 ± 8.58	0.394 ± 0.076	18.7 ± 1.9	33.9 ± 4.4
25 nm −q	166.8 ± 2.05	0.113 ± 0.013	4.63 ± 0.75	10.5 ± 1.6
25 nm N	132.2 ± 6.99	0.294 ± 0.060	28.5 ± 0.26	45.4 ± 4.8
115 nm +q	228.9 ± 5.06	0.326 ± 0.021	–	–
115 nm −q	152.7 ± 1.19	0.140 ± 0.006	–	–
115 nm N	127.2 ± 1.10	0.081 ± 0.015	–	–

Values are mean ± SD (standard deviation) of four independent experiments. PdI: polydispersity index.

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
