# Peer review of "Toxicological Evaluation of SiO2 Nanoparticles by Zebrafish Embryo Toxicity Test"

_ijms, 2019, doi:10.3390/ijms20040882_

Reviewer 1 Report

In this research article, the authors study the toxicity of SiO2 nanoparticles with different sizes and surface charges on zebrafish. The particles were incubated with developing embryos, and the effect of the particles on survival rate, matching time, and morphology were studied. Further, the authors show that the majority of the particles localize to the chorion, and the removal of the the chorion increased the toxicity of positively charged particles. Overall, this work is of high importance and studies the toxicity of commonly occurring nanoparticles in a sensitive biological system. The conclusions could potentially be applied to other types of nanoplatforms in the future.

The authors demonstrate that the nanoparticles are aggregated. Can they comment on how this may affect the toxicity of the particles? Do they feel that using the particles in this aggregated form is representative of real-world scenarios?

Author Response

Point-by-point response to the comments of Reviewer #1

We thank the reviewer for the helpful comments.

In this research article, the authors study the toxicity of SiO2 nanoparticles with different sizes and surface charges on zebrafish. The particles were incubated with developing embryos, and the effect of the particles on survival rate, matching time, and morphology were studied. Further, the authors show that the majority of the particles localize to the chorion, and the removal of the chorion increased the toxicity of positively charged particles. Overall, this work is of high importance and studies the toxicity of commonly occurring nanoparticles in a sensitive biological system. The conclusions could potentially be applied to other types of nanoplatforms in the future.

The authors demonstrate that the nanoparticles are aggregated. Can they comment on how this may affect the toxicity of the particles? Do they feel that using the particles in this aggregated form is representative of real-world scenarios?

Response: We would like to thank the reviewer for this remark. Indeed, the aggregation state of NPs could influence the final toxicological outcome. Baring that in mind we performed series of experiments using NPs dispersed in Danieau’s solution supplemented with albumin (0.012 mg/ml), in order to obtain a better dispersion of NPs. However, this experimental set-up (presence of albumin in Danieau’s solution) was toxic for the embryos on its own, without any presence of NPs. Moreover, dispersibility of NPs was not much improved. Therefore, we decided to continue the treatment of the embryos using media that is the most suitable and destined for their normal growth and development. It is a good idea to compare aggregation state of NPs in Danieau’s solution against freshwater, as a normal habitat for the fish. This could be objective of our follow up paper. We added this information in the Results and Discussion section (page 4, lines 180-189).

Reviewer 2 Report

The manuscript entitled “Toxicological Evaluation of SiO2 Nanoparticles by Zebrafish Embryo Toxicity Test” submitted by Vranic and collaborators describes the evaluation of SIO2 nanoparticles toxicity in zebrafish embryo. They use different nanoparticles concerning the size (25 and 115 nm) and the charge (neutral, negative and positive) to evaluate their toxicity. In their work, the authors assessed the NPs toxicity effects in zebrafish embryos by measuring survival rate, hatching time, gross morphological changes, VEGF and VEGF receptor expression. The manuscript is suggested for publication by International Journal of Molecular Sciences after major revision.

Major issues may be addressed as listed below:

1- In Figure 1b (Hatching time), the SD values for the Nps of 25 nm (+q) at 48 hpf and for Nps 25 nm (-q) at 72 hpf are too high rendering the interpretation of the NPs on hatching time not accurate.

2- In Figure 2, the magnification of the images is not enough to observr clearly if there are any morphological changes.

3- In Figure 3, again the SD values are too high to conclude on the NPs effects in the gene expression of VEGF and its receptor.

4- There is confusion about the concentration of NPs, the authors use mg/mL and sometimes mg/dL. For clarity, I suggest to uniform these concentration units.

5- It is intriguing to still have the chorion at 72 hpf in Figure 4. At this stage of development, the chorion is lost and the embryos are free.

6- The experiments in the absence of chorion are not shown. I suggest to add the results of these experiments which are important to evaluate definitively the toxicity of the NPs since the chorion disappear after 48 hpf.

7- It is interesting to study the toxic effects of NP when the larva has become independent after the nutrients contained in the yolk sac. are exhausted.

From all of these points, I consider that this manuscript is suitable for publication after major revision.

Author Response

Point-by-point response to the comments of Reviewer #2

We thank the reviewer for helpful comments and remarks that helped us improve the manuscript significantly. We took all the comments into account and addressed them point by point below.

The manuscript entitled “Toxicological Evaluation of SiO2 Nanoparticles by Zebrafish Embryo Toxicity Test” submitted by Vranic and collaborators describes the evaluation of SiO2 nanoparticles toxicity in zebrafish embryo. They use different nanoparticles concerning the size (25 and 115 nm) and the charge (neutral, negative and positive) to evaluate their toxicity. In their work, the authors assessed the NPs toxicity effects in zebrafish embryos by measuring survival rate, hatching time, gross morphological changes, VEGF and VEGF receptor expression. The manuscript is suggested for publication by International Journal of Molecular Sciences after major revision.

Major issues may be addressed as listed below:

1. In Figure 1b (Hatching time), the SD values for the NPs of 25 nm (+q) at 48 hpf and for NPs 25 nm (-q) at 72 hpf are too high rendering the interpretation of the NPs on hatching time not accurate.

Response: We have now revised and pooled our data from several experiments in order to increase the number of analyzed eggs for 25 nm, -q condition. Indeed, we have not observed such a huge variation in hatching pattern of the fish in previous experiments. When looking more carefully, we also noticed that out of three experiments, in two of them the difference in % of hatched fish comparing to the untreated fish was not significantly different, while in one experiment it was lower. We believe our data reflect that trend more clearly now. As for the 25 nm, +q NPs at 48 hpf we unfortunately discovered a mistake in our calculations, we corrected this mistake by recalculating the % of hatched fish at this time point. We sincerely apologize for this mistake. All taken together, we replaced the Figure 1b with the new data.

2. In Figure 2, the magnification of the images is not enough to observe clearly if there are any morphological changes.

Response: We would like to thank the reviewer for this remark. We have now replaced all images in Figure 2 using better resolution images. We still think that it is more representative to show the effect of NPs using higher number of the fish in the dish. However, we do focus on one/few fish in insets and show in a zoomed manner their appearance in order to demonstrate in a better detail that “no effect” of NPs on the fish morphology was observed.

3. In Figure 3, again the SD values are too high to conclude on the NPs effects in the gene expression of VEGF and its receptor.

Response: We thank the reviewer for the comment. As pointed out by the reviewer, the SD values are relatively high in Figure 3 that shows gene expression of VEGF and its receptors. However, these variations is possible because another paper [#28] that investigated the expression of VEGF in zebrafish embryos similarly showed the high SD in their data. Total RNA was extracted from the whole body of zebrafish and was subjected to quantitative PCR in both studies. That might explain how the variations of gene expression of VEGF and its receptors were high. If the expression of VEGF and its receptors was analyzed using total RNA isolated from the sorted GFP-positive endothelial cells, the variations might be small. Further studies are needed to evaluate whether SiO2 NPs effect on angiogenesis and the gene expression associated with angiogenesis. We added the discussion in the Results and Discussion section (page 7, lines 236-244).

4. There is confusion about the concentration of NPs, the authors use mg/mL and sometimes mg/dL. For clarity, I suggest to uniform these concentration units.

Response: We apologize for the typo in some Figure captions. Now it has been corrected, since in all the experiments the metrics we used were expressed as mg/L.

5. It is intriguing to still have the chorion at 72 hpf in Figure 4. At this stage of development, the chorion is lost and the embryos are free.

Response: As shown in our Figure 1b, at 72 hpf also for the untreated fish, even though the majority of the fish has hatched, there was small number of the fish still having chorion. However, since the majority of the fish were hatched we agree with the reviewer that showing hatched fish is definitely more representative. Therefore, we replaced image of the untreated fish at 72 hpf in Figure 4a.

6. The experiments in the absence of chorion are not shown. I suggest to add the results of these experiments which are important to evaluate definitively the toxicity of the NPs since the chorion disappear after 48 hpf.

Response: In the preset study, we focused on the effects of SiO2 NPs on the development of zebrafish and also the effects of SiO2 NPs on angiogenesis. Because aorta and large vessels are formed by 48 hpf and small vessels in abdomen such as subintestinal vessels are formed by around 72 hpf in zebrafish [#43], we exposed SiO2 NPs at 6 hph and observed the effect of SiO2 NPs on the formation of blood vessel at 72 hpf. As pointed out by the reviewer, the evaluation of the toxicity of SiO2 NPs by the exposure to SiO2 NPs after the chorions disappear at 48 hpf are also important. We need further investigation to determine the effects of SiO2 NPs after the absence of chorion. We added the discussion in the Results and Discussion section (page 9, lines 296-301).

7. It is interesting to study the toxic effects of NP when the larva has become independent after the nutrients contained in the yolk sac are exhausted.

Response: As pointed out by the reviewer, it is also interesting to evaluate the effects of NPs on young zebrafish (after 8 dpf) because the larvae become independent and start oral feeding after the nutrients contained in the yolk sac are exhausted. This topic can be considered as the essential theme of further investigation. We added this information in the Results and Discussion section (page 9, lines 301-303).

Round  2

Reviewer 2 Report

The manuscript entitled “Toxicological Evaluation of SiO2 Nanoparticles by Zebrafish Embryo Toxicity Test” submitted by Vranic and collaborators describes the evaluation of SiO2 nanoparticles toxicity in zebrafish embryo. The authors answered to all the comments and the manuscript has been significantly improved-It is now suitable for publication in the IJMS.